# Food Webs and Fish Size Patterns in Insular Lakes Partially Support Climate-Related Features in Continental Lakes



Nicolas Vidal [1,2,3,*], Susanne L. Amsinck [1], Vítor Gonçalves [4,5], José M. Neto Azevedo [4], Liselotte S. Johansson [1], Kirsten S. Christoffersen [6], Torben L. Lauridsen [1,7], Martin Søndergaard [1], Rikke Bjerring [1], Frank Landkildehus [1], Klaus P. Brodersen [6], Mariana Meerhoff [1,3] and Erik Jeppesen [1,2,8,9]

1 Department of Bioscience, Aarhus University, Vejlsøvej 25, 8600 Silkeborg, Denmark; sla@dmu.dk (S.L.A.); lk07@science.au.dk (L.S.J.); tll@bios.au.dk (T.L.L.); ms@bios.au.dk (M.S.); rbh@clin.au.dk (R.B.); fla@bios.au.dk (F.L.); mm@bios.au.dk (M.M.); ej@bios.au.dk (E.J.)
2 Sino-Danish Centre for Education and Research (SDC), University of Chinese Academy of Sciences, Beijing 100190, China
3 Departamento de Ecología y Gestión Ambiental, CURE, Universidad de la República, Maldonado 20100, Uruguay
4 Faculty of Sciences and Technology, University of the Azores, R. Mãe de Deus, 9500-321 Ponta Delgada, Azores, Portugal; vitor.mc.goncalves@uac.pt (V.G.); josemvazevedo@uac.pt (J.M.N.A.)
5 CIBIO, Centro de Investigação em Biodiversidade e Recursos Genéticos, InBIO Laboratoório Associado, Pólo dos Açores, Rua Mãe de Deus 13 A, 9500-321 Ponta Delgada, Azores, Portugal
6 Department of Biology, University of Copenhagen, Universitetsparken 4, 2100 Copenhagen, Denmark; kchristoffersen@bio.ku.dk (K.S.C.); klaus.brodersen@gmail.com (K.P.B.)
7 Arctic Research Centre, Aarhus University, C.F. Møllers Allé 8, Bldg. 1110, 8000 Aarhus C, Denmark
8 Limnology Laboratory, Department of Biological Sciences and Centre for Ecosystem Research and Implementation, Middle East Technical University, Ankara 06800, Turkey
9 Institute of Marine Sciences, Middle East Technical University, Erdemli-Mersin 33731, Turkey
* Correspondence: ocin101@hotmail.com; Tel.: +598-99-709-679

**Abstract:** Disentangling the effects of climate change on nature is one of the main challenges facing ecologists nowadays. Warmer climates forces strong effects on lake biota for fish, leading to a reduction in size, changes in diet, more frequent reproduction, and stronger cascading effects. Space-for-time substitution studies (SFTS) are often used to unravel climate effects on lakes biota; however, results from continental lakes are potentially confounded by biogeographical and evolutionary differences, also leading to an overall higher fish species richness in warm lakes. Such differences may not be found in lakes on remote islands, where natural fish free lakes have been subjected to stocking only during the past few hundred years. We studied 20 species-poor lakes located in two remote island groups with contrasting climates, but similar seasonality: the Faroe Islands (cold; 6.5 ± 2.8 °C annual average (SD) and the Azores Islands (warm; 17.3 ± 2.9 °C)). As for mainland lakes, mean body size of fish in the warmer lakes were smaller overall, and phytoplankton per unit of phosphorus higher. The $\delta^{13}$C carbon range for basal organisms, and for the whole food web, appeared wider in colder lakes. In contrast to previous works in continental fresh waters, Layman metrics of the fish food web were similar between the two climatic regions. Our results from insular systems provide further evidence that ambient temperatures, at least partially, drive the changes in fish size structure and the cascading effects found along latitude gradients in lakes.

**Keywords:** stable isotopes; climate change; islands; space-for-time substitution

## 1. Introduction

Unraveling the effects of climate change and its impact on natural systems has been a central goal in ecological studies, being particularly important in the context of global change [1,2]. Numerous approaches have been used to study the effects of climate change in shallow lakes [3], including warming experiments [4–6], mathematical models [7],

paleolimnological records [8], time series [9], and space-for-time substitution (SFTS) [10]. In particular, SFTS has provided major empirical evidence of climate change effects in lakes, including shallow lake systems [3,10]. In this approach, comparisons were conducted between systems in contrasting climates where the organisms had enough time to evolve and adapt to the local climate. However, apparent climate effects determined by SFTS may in part reflect differences on biogeography, and may be influenced by co-variates of temperature and the latitudinal variation in other climate-related variables, such as seasonality and the length of the growing season. Similar studies focusing on species-poor areas, such as remote islands located in contrasting climates with a recent fish introduction, may be useful to somewhat separate the effect of confounding factors occurring at a large, latitudinal scale in the mainland [3].

Studies conducted along climate gradients suggest strong effects of increases in ambient temperatures on lake fish community structures and dynamics, as reviewed in [3]. Reduced abundance of cold stenothermal fish species and enhanced abundance of eurythermal fish species, even in deep, stratified lakes, are expected with decreasing latitude [11]. Moreover, fish in warmer climates have higher specific metabolic and excretion rates, as well as earlier reproduction and prolonged spawning [12–14]. Moreover, a decrease in mean fish body size [15–20] is evident with decreasing latitude and have been attributed particularly to a significant switch in life history [19], and sometimes to intra-species density depended changes [21]. Moreover an increase in taxonomic and functional richness [16–18,22] as well as changes in functional composition with increased proportion of omnivorous and herbivorous fish species [23,24], have been observed with decreasing latitude.

Due to reduced mean size, among other expected changes in fish community structure and function at warmer regions, a higher zooplanktivory is expected [25–27]. The higher fish predation pressure on zooplankton reduces the grazing pressure on phytoplankton [10] and the maintenance of clear water at the ecosystem level is consequently less frequent in warm than in cold climates [28].

The described differences at community level can affect various food web metrics, including the trophic web length (TWL), commonly measured as range in $\delta^{15}$N (NR). TWL has been one of the most studied metrics of trophic webs and several hypotheses and mechanisms have been proposed to explain its variations. One is the energetic limitation hypothesis (ELH) [29–31]; and references therein, according to which energy losses between trophic levels limit the maximum food web length, is susceptible to temperature changes. A negative correlation between the maximum trophic position and environmental temperature is theoretically expected [32], a pattern already revealed in empirical studies [33–35].

The width of the food web, represented as $\delta^{13}$C range (hereafter CR), depends largely on the stable isotope signature of primary producers [36], but as their signal is highly variable among ecosystems, primary consumer's signatures have been proposed to better integrate the natural variability in space and time of the basal resources (producers) [36]. The carbon signal ($\delta^{13}$C) was thought to be transmitted with little change from the primary producers to the consumers; consequently, the consumer signals depend directly on the signals of their diets [37]. However, a large variability in trophic fractionations has been reported for aquatic consumers [38]. In contrast to nitrogen, the theoretical framework developed for carbon is patchier, particularly in connection with climate and temperature in particular [39]. In low productivity lakes, the carbon sources for photosynthesis usually derive from mineralization in the benthic and littoral zones, which are typically depleted in $\delta^{13}$C compared to atmospheric carbon signature [40]. In productive lakes, however, endogenous carbon may not suffice to maintain pelagic photosynthesis and pelagic producers therefore have to exploit a higher proportion of atmospheric inorganic carbon, enriched in $\delta^{13}$C [40,41]. An enriched $\delta^{13}$C signal, due to input of atmospheric carbon in the pelagic zone in the warmer (and consequently, often more productive) lakes is thus expected, which may reduce the range of $\delta^{13}$C of producers in warmer systems.

Biotic interactions could also lead to differences in CR as fish predation may affect the abundance, composition, size structure, and/or the diet behavior of macroinvertebrates [42–44], and consequently reduce their use of resources. Accordingly, lake food webs with stronger fish predation, namely warm low latitude systems, would potentially exhibit a narrower CR; this has previously been observed in streams [45].

To reduce some of the weakness of SFTS performed in continental lakes, often positioned in distant regions with different seasonality, biogeography, and evolutionary time [3], we studied two sets of species-poor lakes located in two remote groups of islands in the North Atlantic Ocean with contrasting climates, but comparable seasonality, namely the Azores (warm) and the Faroe (cold) Islands. Since both island groups are located in the North Atlantic Ocean, the studied lakes are subject to the same seasonality (Figure 1).

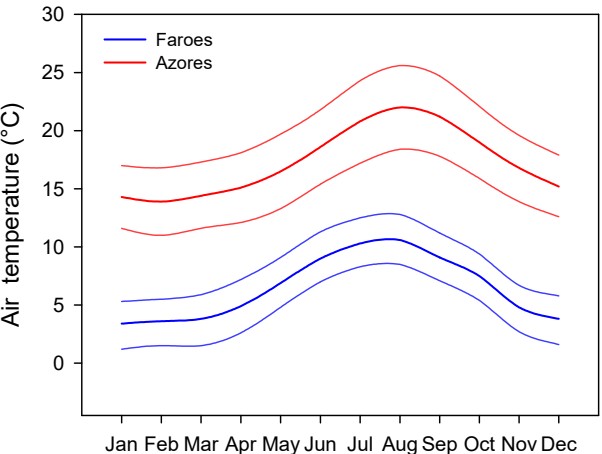

**Figure 1.** Average air temperature as full lines (average daily temperature per month) and maximum and minimum temperatures as dotted lines (average daily temperature per month) for Azores (red) and Faroe Islands (blue) during the period 1961–1990. Source: https://en.climate-data.org/europe/ (accessed on 13 May 2021).

Records of freshwater fish communities in both sets of islands trace back to the 17th and 18th centuries [46,47]. Fish were first introduced by humans in the Azores Islands in 1792 [47], while the fish species at the Faroe Island are indigenous, but have been transplanted internally [46,48]. Most of the fish species in the Azores Islands were stocked for the first time one to two hundred years ago. The first introduced species in 1792 was crucian carp *Carassius auratus* Linnaeus 1758. Most of the exotic species were introduced during the 19th century; ruivaca *Achondrostoma oligolepis* (Robalo, Doadrio, Almada and Kottelat 2005) was introduced in 1879; brown trout *Salmo trutta* Linnaeus 1758 in 1880; common carp *Cyprinus carpio* Linnaeus 1758 in 1890; roach *Rutilus rutilus* (Linnaeus 1758) in 1895; largemouth black bass *Micropterus salmoides* (Lacepède 1802) in 1898 and perch *Perca fluviatilis* Linnaeus 1758 in 1898. Three species have been more recently stocked, rainbow trout *Oncorhynchus mykiss* (Walbaum 1792) in 1941, pike *Esox Lucius* Linnaeus 1758 in 1979 and finally pikeperch *Sander lucioperca* (Linnaeus 1758) in 1980.

We used a stable isotope approach to test the hypothesis that ambient temperature differences between the two island groups play a key role in shaping food webs. Based on evidence from mainland lakes, we expected to find: (1) a fish size structure dominated by small individuals in the warmer lakes (Azores Island); (2) a stronger top-down effect of fish in the warmer lakes than in the colder ones (Faroe Island); (3) shorter food webs in the warmer lakes; and (4) a wider (CR) food web in the colder lakes.

## 2. Materials and Methods

### 2.1. Study Sites

The Azores Islands are a group of nine volcanic islands located 1500 km west off the coast of Portugal and 850 km northwest from the island of Madeira in the North Atlantic Ocean (between 37°44′ and 37°44′ N and between 25°08′ and 25°51′ W). The Faroe Islands are a treeless archipelago also volcanic origin composed of 18 islands located in the North Atlantic Ocean (between 61°20′ and 62°24′ N and between 6°15′ and 7°41′ W), their nearest neighbors being the Shetland Islands (345 km), Iceland (430 km), and Norway (575 km) (Figure 2). Annual mean air temperatures are higher in the Azores ($17.3 \pm 2.9$ °C) than in the Faroe Islands ($6.5 \pm 2.8$ °C) ($t_{1,22} = 9.37$, $p < 0.001$). Moreover, the temperature difference between regions is relatively constant along the year, ranging from 9.5 to 12.8 °C (Figure 2). The lowest temperatures are recorded between January and February (11 °C in the Azores and 1.5 °C in the Faroe Islands) and maximum temperatures occur around August (25.6 °C in the Azores and 12.8 °C in the Faroe Islands; source: https://en.climate-data.org/europe/) (accessed on 13 May 2021). The lake area (ha) of the studied lake was estimated using a geo-referenced satellite image using the free software Quantum-GIS 2.6.1.

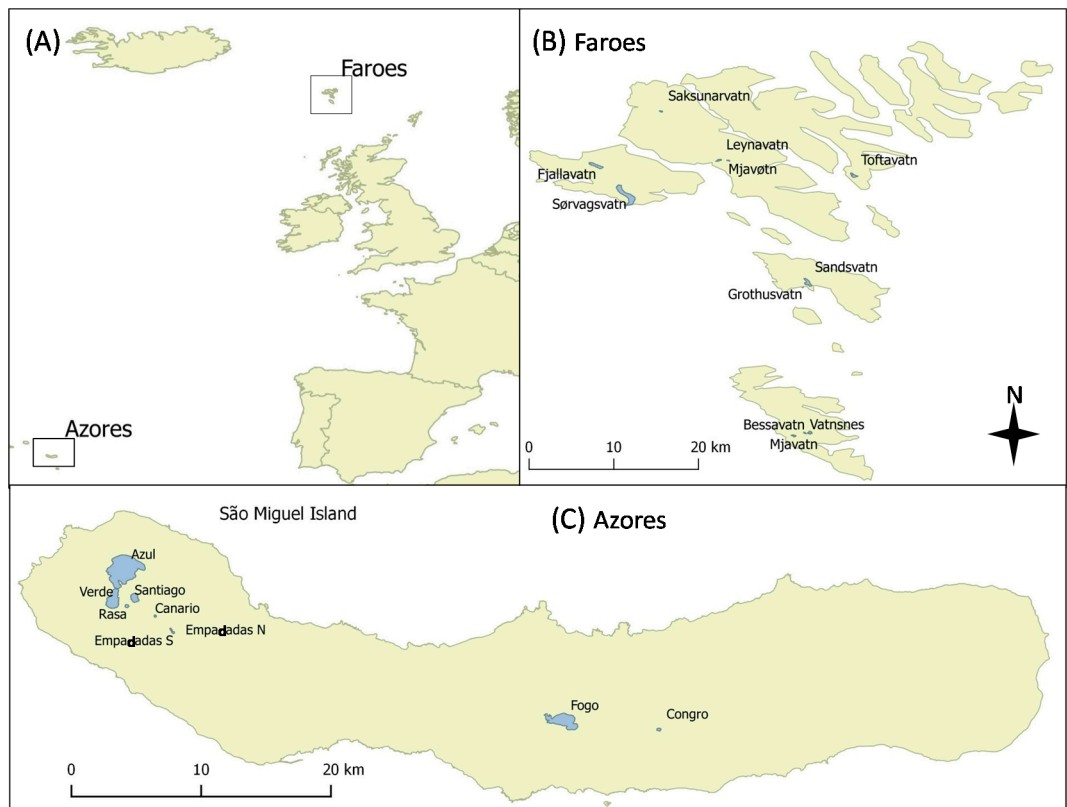

**Figure 2.** Location of the two groups of island (**A**), showing the studied lakes in the Faroe Islands (**B**) and the Azores, São Miguel Island (**C**).

### 2.2. Sampling

We sampled 9 lakes in the Azores and 11 in the Faroe Islands during summer 2006 and 2000, respectively. Depth-integrated water samples were taken at the deepest point of each lake (see Table 1) and preserved for subsequent laboratory determination of nutrient and phytoplankton chlorophyll-*a* concentration (Chl-*a*). Profiles of lake water temperature and dissolved oxygen (when the lakes were stratified), pH, and conductivity were recorded using an Hydrolab, Surveyor 4 (Hach Environmental, Loveland, CO, USA) and Secchi depth was measured. Additionally, a depth-integrated sample for zooplankton (~20 L) was taken with a Schindler-Patalas Plankton Trap at the deepest point of the lake (integrating

several samples from surface to bottom), and the water filtered through a 20-μm mesh net, back washed into filtered water and fixed with 4% Lugol's solution. The deepest point of each lake was determined using a portable echo sounder (Hondex ps-7, Toyohashi, Japan). In the laboratory, the zooplankton individuals were identified to species level and counted using an inverted light microscope (Leitz Labovert FS, Ernst LeitzLtd., Midland, ON, Canada)). A subsample of 20–30 randomly selected individuals per species was measured (to 0.01 mm) using an ocular scale and biomass was calculated using density of organisms and published length–weight relationships [49].

**Table 1.** Morphometric, basic physical, chemical, and biological characteristics of the study lakes ordered by increasing mean fish size.

| Island | ID | Lake Name | A | $Z_{max}$ | S | pH | TN | TP | Chl-*a* | Zoo | Zoo | Fish Size |
|---|---|---|---|---|---|---|---|---|---|---|---|---|
| | | | (ha) | (m) | (m) | | ($\mu g\ L^{-1}$) | ($\mu g\ L^{-1}$) | ($\mu g\ L^{-1}$) | ($\mu g\ L^{-1}$) | (ind. $L^{-1}$) | (cm) |
| Azores Islands | A1 | Empadadas N. | 1.8 | 6 | 0.72 | - | 710 | 33 | 75.7 | 177.8 | 135.3 | 6.9 |
| Azores Islands | A2 | Canario | 1.8 | 2.3 | 1.8 | 7.3 | 440 | 13 | 6.7 | 7.5 | 5.1 | 7.9 |
| Azores Islands | A3 | Congro | 5.1 | 18.9 | 2.4 | 7.1 | 530 | 20 | 5.6 | 85.2 | 61.2 | 9.6 |
| Azores Islands | A4 | Empadadas S. | 1.5 | 3.0 | - | 7.3 | 350 | 11 | 3.8 | 1.7 | 1.3 | 10.1 |
| Azores Islands | A5 | Verde | 86.3 | 24.0 | 2.7 | 8.1 | 330 | 29 | 6.5 | 123.5 | 68.1 | 12.0 |
| Azores Islands | A6 | Azul | 361 | 28.5 | 2 | 6.9 | 900 | 23 | 5.7 | 52.6 | 23.1 | 13.6 |
| Azores Islands | A7 | Santiago | 25.4 | 28.8 | 5.2 | 8.6 | 260 | 13 | 3.6 | 40.0 | 13.7 | 17.3 |
| Azores Islands | A8 | Fogo | 153.0 | 31.6 | 2.7 | 6.6 | 190 | 13 | 3.8 | 22.8 | 11.7 | 17.6 |
| Azores Islands | A9 | Rasa SC | 3.9 | 4.5 | - | 7.2 | 190 | 4 | 2.0 | 16.0 | 1.5 | 17.6 |
| Faroe Islands | F1 | Sørvágsvatn | 356.0 | 52.0 | 12.5 | 6.9 | 318 | 5 | 0.7 | 23.9 | 5.2 | 13.4 |
| Faroe Islands | F2 | Vatnsnes | 14.7 | 9.5 | 1.7 | 6.6 | 780 | 76 | 25.2 | 174.0 | 42.5 | 17.4 |
| Faroe Islands | F3 | Leynavatn | 18.0 | 32.5 | 10 | 6.9 | 168 | 3 | 1.2 | 32.9 | 21.7 | 17.2 |
| Faroe Islands | F4 | Toftavatn | 52.2 | 17.5 | 5.8 | 6.8 | 220 | 11 | 1.0 | 126.5 | 25.6 | 18.4 |
| Faroe Islands | F5 | Fjallavatn | 101.9 | 46.6 | 14 | 6.6 | 100 | 3 | 0.5 | 40.4 | 4.8 | 18.7 |
| Faroe Islands | F6 | Mjáuvøtn | 3.1 | 5.7 | 4.3 | 6.8 | 252 | 15 | 1.8 | 874.7 | 175.5 | 19.8 |
| Faroe Islands | F7 | Gróthúsvatn | 13.4 | 0.7 | 0.7 | 8.8 | 500 | 35 | 1.0 | 154.3 | 22.8 | 23.4 |
| Faroe Islands | F8 | Mjávavatn | 0.6 | 0.8 | 0.8 | 7.0 | 250 | 19 | 1.8 | - | - | 25.2 |
| Faroe Islands | F9 | Saksunarvatn | 8.1 | 16 | 8.8 | 8.0 | 116 | 6 | 1.1 | 476.1 | 53.2 | 25.4 |
| Faroe Islands | F10 | Sandsvatn | 79.7 | 2.4 | 2.4 | 7.7 | 310 | 43 | 1.1 | 357.7 | 85.6 | 27.1 |
| Faroe Islands | F11 | Bessavatn | 5.4 | 2.0 | 2.0 | 6.8 | 250 | 30 | 2.0 | 371.5 | 54.0 | 28.7 |

A = lake area; $Z_{max}$ = maximum depth; S = Secchi depth; TN = total nitrogen; TP = total phosphorus; Chl-*a* = chlorophyll-a concentration; zoo, crustacean zooplankton, fish size, total length of fish.

Chl-*a* concentration was used as a proxy of phytoplankton biomass. For the analysis, 100–1000 mL water, depending on the concentration, were filtered in the field through Whatman GF/C filters (47 mm in diameter), kept dark and frozen within few hours, followed by spectrophotometric analysis after ethanol extraction in the laboratory [50]. Total phosphorus (TP) was determined as molybdate reactive phosphorus [51], following persulfate digestion [52] and total nitrogen (TN) was determined as nitrite after potassium persulfate digestion [53].

Fish were captured using Lundgren gillnets with 14 mesh sizes ranging from 6.25 to 75.0 mm from knot-to-knot, as well as by fyke nets to supplement the capture of species that were not easily caught in gillnets (e.g., eel, *Anguilla* spp.). The fyke nets [54] used had a central leader 8 m long, and a height of 0.6 m, two internal valves (0.55 m); nets were made throughout of 8 mm mesh (knot-to-knot), except the end of the funnel was 4 mm; no bait was placed in the nets. The number of gillnets used depended on lake size and ranged from 2 to 6 nets per lake. The gillnets were placed overnight in the littoral and pelagic zone. If shallow (mean depth < 3 m) only benthic nets were used and half of the nets were set in each zone; if mean depth was >3 m 1/3 of the nets were set as for shallow lakes and an additional 1/3 of the nets were pelagic nets positioned in the middle of the epilimnion. Moreover, one fyke net was placed near the shore. Fish density was calculated as CPUE (capture per unit effort, number of fish $net^{-1}\ h^{-1}$) and BPUE (biomass per unit effort, grams of fish $net^{-1}\ h^{-1}$) [55]. All fish were weighed (TW, 0.01 g) and total length was measured (TL, 0.1 cm), and a representative subsample of the capture and sizes of each species was used for stable isotope analyses. Fish muscle samples were taken and frozen (−18 °C) immediately until analysis in laboratory.

For stable isotope analysis, we further collected samples of: (1) each macrophyte species present (collected by hand); (2) three littoral periphyton samples (collected by scraping rocks from the shore using a brush); (3) three benthic periphyton samples (as

the top 0.5 cm slice of sediment cores collected using a Kajak corer in three regions of the lake); (4) three seston samples (i.e., matter retained on a 11 µm-mesh net after pre-filtration through a 80 µm-mesh net to remove zooplankton of lake water collected in three different zones of each lakes); (5) three zooplankton samples (using vertical nets with two mesh sizes, 140 µm and 500 µm, to include organisms of different sizes); (6) littoral macroinvertebrates (using a kick-net in the littoral zone); and (7) macroinvertebrates from the deep zone (using a benthic sledge). In lakes where macrophytes were abundant, also macroinvertebrates in the macrophyte beds were collected with a sweep net. When possible, we collected three replicates of each item, the samples were frozen immediately (−18 °C) until analysis in laboratory.

Samples for stable isotope analyses were processed separately in the laboratory to avoid contamination. The samples were freeze dried and ground into a fine powder that was kept in glass vials and then weighed to 0.01 mg precision using an analytical scale (Sartorius ME235S Genius, Sartorius, Göttingen, Germany). The samples were then loaded into tin capsules and forwarded to UC Davis Stable Isotope Facility (University of California, Davis, CA, USA) for carbon and nitrogen stable isotopes analysis. Lipids were not chemically extracted because the protocols used to remove lipids may affect nitrogen isotope integrity [56,57], whereas at the same time, when the C:N ratio of samples was low (C:N < 3.5 for aquatic animals) extraction or normalization has insignificant influence on the d$^{13}$C values [56]. Since in most fish individuals analyzed here (~80%) the C:N was <3.5, lipid concentrations should be uniformly low; and consequently, we used the raw isotopic data. The trophic position (based on $\delta^{15}$N values, hereafter TPos), and $\delta^{13}$C values for all available organisms were plotted to reconstruct the food web structure of each lake [58]. The trophic position of each individual was determined according to [59]:

$$\text{Trophic position of consumers} = [(\delta^{15}N_{consumers} - \delta^{15}N_{base})/2.98] + 2 \qquad (1)$$

where $\delta^{15}N_{consumers}$ is the isotopic signature of each individual analyzed and $\delta^{15}N_{basal}$ corresponds to the herbivorous invertebrates with the lowest $\delta^{15}$N value, 2.98 is the expected $\delta^{15}$N fractionation per trophic level [60], and 2 is the theoretical trophic level of the basal organisms.

### 2.3. Data Analysis

The physical and chemical parameters and morphometric characteristics of the lakes from the two climatic regions were compared using the Mann–Whitney U-test. The potential maximum colonization depth ($C_{max}$) of submerged macrophytes was calculated using [Log($C_{max}$) = 0.32 − 0.23 × log(TP) − 0.15 × log(Chl-*a*)] based on [61], as a potential indicator of benthic and littoral heterogeneity provided by macrophytes.

To test the hypothesis regarding the size structure of fish in the two climatic regions, the mean total length frequency in each size class (cm) was calculated and plotted for each region. In addition, the mean size of fish in each lake was compared between the two climatic regions using parametric T-tests. As all the fish species in the Azores Islands have been stocked and therefore by chance might have a smaller size than those present in the lakes in the Faroe Islands, we also compared the size of fish relative to the maximum size for the species reported in the Fishbase [62]. For anadromous species we used the maximum size reported for freshwater resident forms (see Table S3 in Supporting Information), since marine forms are typically larger, and all the studied lakes had only resident populations. To mitigate additional potentially confounding factors, we also compared the size of the fish caught excluding juveniles, determined from maturity sizes reported in the literature (Table S5). Since *O. mykiss* is stocked every year by the Azorean Forest Department (José Manuel Neto Azevedo, pers. comm.), it was excluded from the size analysis.

To test the hypothesis of stronger top-down effect of fish in the warmer lakes, we calculated several indicators of potential fish predation pressure and top-down effects on plankton communities, i.e., fish:zooplankton biomass ratio, zooplankton mean size (mm), cladoceran mean size (mm), total zooplankton biomass (µg L$^{-1}$), total zooplankton:

phytoplankton biomass ratio, Chl-*a*: total phosphorus ratio, and Chl-*a*: total nitrogen ratio [63]. All of these indicators, as well as macroinvertebrate composition, were compared using parametric T-tests. We calculated the percentage of small fish (<10 cm TL) in each lake (hereafter %SF) and performed linear regressions between %SF and potential predation pressure indicators. ANCOVA was used to test for differences in Chl-*a* concentrations between both regions, using TP as co-variable. Statistical significance was established at the 0.05 level, and all the analyses were performed using the *aov* function in R.

To test the hypotheses regarding the food web metrics, the trophic structure of the basal organisms, fish community, and the whole food web were described using the community-wide metrics of [36], applying the package SIBER in R [64]. These metrics include the total area (TA) calculated as the convex hull of the food web, and represents trophic diversity; the nitrogen range (NR) that represents trophic height, and the carbon range (CR) that represents the width of the food web. The latter two were calculated as the difference between the maximum and minimum values. Since consumer CR and NR can be affected by basal CR and NR lake variability due to intrinsic factors [65], we also calculated the CR of fish standardized by the CR of the baseline organisms and trophic position, in order to compare between region differences. The centroid distance (CD) is calculated as the average Euclidean distance of each element to the centroid, and represents the average trophic diversity. Finally, the mean nearest neighbor distance (MNND) and standard deviation of the nearest neighbor distance (SDNND) are related to trophic redundancy. The Layman metrics were calculated separately for basal organisms (using littoral and benthic invertebrates and pelagic crustaceans), and for fish, applying a balanced randomly selected number of individuals (i.e., for rare spp. with n < 15, we used all individuals, and for the more abundant ones, we randomly picked 15 individuals per species), as some of these metrics are sensitive to the number of samples [66]. Layman metrics were also calculated for the entire food web (combining basal organisms, other consumers, and fishes). Since the total area (TA) metric is affected by extreme values and, therefore, increases with sample size, we also calculated the standard ellipses areas with and without correction for sample size (SEA$_c$) and (SEA) [64], respectively, to compare the food webs between regions.

All of the trophic metrics were compared using the Mann–Whitney test, while the trophic position was compared using T-test. The relationship between the trophic position of the fish and basal organisms was analyzed using linear regression models. To analyze the shape of the lake food web in each region, we calculated the difference between CR at the base and top of the food web, and were compared between regions using T-test. The potential causes of the differences in the CR of basal organisms between regions were analyzed using linear regression models, with environmental variables (lake area, maximum depth, Secchi depth, Secchi depth:maximum depth ratio, total phosphorus, total nitrogen, Chl-*a*, potential maximum colonization depth of submerged macrophytes (C$_{max}$), pH, conductivity, $\delta^{13}$C signal of pelagic zone, CR of primary producers), and predator indicators (total zooplankton:phytoplankton biomass ratio, fish:zooplankton biomass ratio, % small fish (<10 cm TL), CPUE small fish) as explanatory variables. All of the variables were analyzed using parametric tests when the assumptions were met, otherwise non-parametric tests were used.

## 3. Results

### 3.1. Environmental Characteristics

The two sets of lakes were selected in order to include a wide and similar range of environmental variables (Table 1). Accordingly, no statistically significant differences in morphometric variables were detected between the regions (Table 2). However, morphometric characteristics varied considerably among the lakes, with lake area ranging from 0.6 to 356 ha in the Faroe Islands and from 1.8 to 392 ha in the Azores, and maximum depth from 0.7 to 52 m in Faroese and between 2.5 and 32.3 m in Azorean lakes (Table 1). Moreover, nutrient concentrations varied across lakes, but no statistically significant differences were detected between regions, except for mean Chl-*a* concentration,

being higher in the Azorean than in Faroese lakes (Table 2), even after removing differences in TP concentrations (ANCOVA $F_{(1-17)}$ = 10.9, $p < 0.05$). There were no significant differences in mean water transparency (with Secchi depth ranging from 0.7 to 12.5 m in the Faroese lakes and from 0.7 to 5.2 m in the Azorean lakes). However, the Secchi depth: maximum lake depth ratio was significantly higher in Faroese lakes, in agreement with the Chl-*a* data (Table 2). Chl-*a* was positively correlated with TP concentrations in Azores (y = 1.16x − 0.58; $R^2$ = 0.52; $p < 0.05$) and Faroe Islands (y = 0.64x − 0.55; $R^2$ = 0.45; $p < 0.05$) (Figure 3A). Chl-*a* was also positively correlated with TN concentrations in both climatic regions, but the regression was only significant in Azorean lakes (y = 1.46x − 3.18; $R^2$ = 0.50; $p < 0.05$) (Figure 3B).

**Table 2.** Statistical comparison of the main lake characteristics between the two regions, indicating mean ± SE and results of statistical tests (Mann–Whitney tests).

|  | Azores Islands | Faroe Islands | Z | *p* |
|---|---|---|---|---|
| Area (ha) | 71 ± 40 | 55 ± 38 | −0.38 | 0.69 |
| Maximum depth (m) | 16 ± 4 | 14 ± 6 | −0.46 | 0.66 |
| pH | 7.4 ± 0.7 | 7.2 ± 0.7 | −1.12 | 0.26 |
| Total Nitrogen (µg L$^{-1}$) | 433 ± 81 | 316 ± 63 | −1.60 | 0.11 |
| Total Phosphorus (µg L$^{-1}$) | 18 ± 3 | 24 ± 8 | −0.08 | 0.94 |
| Secchi depth (m) | 3 ± 1 | 5 ± 1 | −0.91 | 0.36 |
| Secchi depth:maximum depth | 0.2 ± 0.1 | 0.6 ± 0.1 | −2.77 | **0.003** |
| Chlorophyll *a* (µg L$^{-1}$) | 13 ± 8 | 4 ± 3 | −3.12 | **0.001** |
| $\delta^{13}$C signal of pelagic zone | −23.1 ± 2.6 | −27.9 ± 2.8 | −2.78 | **0.003** |
| Total zooplankton density (ind L$^{-1}$) | 36 ± 15 | 49 ± 18 | −0.46 | 0.65 |

Statistically significant regressions given in bold.

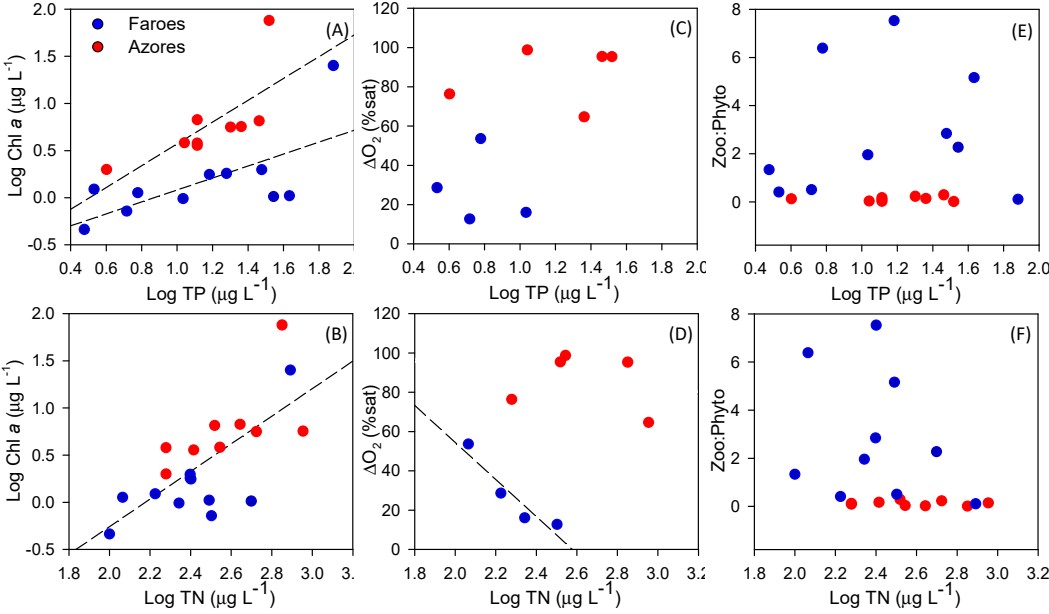

**Figure 3.** Phytoplankton biomass (as log Chl-*a*) versus total phosphorus concentration (as log TP) (**A**), and total nitrogen concentration (as log TN) (**B**); oxygen depletion (ΔO$_2$ %sat) versus log TP in the subset of stratified lakes (**C**) and log TN (**D**); total zooplankton:phytoplankton biomass ratio (zoo:phytobiomass) against log TP (**E**) and log TN (**F**). Red: Azorean lakes, blue: Faroese lakes.

In the Azores Islands, 5 of 9 lakes were stratified and had oxygen depletion in the deeper part (7.7 ± 14.6% saturation), while the surface layers were close to saturation (93.8 ± 9.1% saturation). The bottom oxygen concentration in the stratified Faroese lakes (4 of 11), in contrast, was never critical for the biota (86.4 ± 16.2% saturation). The difference between surface and bottom oxygen, calculated as saturation %, was higher in the Azorean lakes (Figure 3C,D). When analyzing only the stratified lakes in Azores (n = 5) and Faroe

Islands (n = 4), we found higher TP concentration in the warmer (19.6 $\pm$ 6.8 µg TP L$^{-1}$) than in the colder lakes (6.4 $\pm$ 3.2 µg TP L$^{-1}$), (Z = $-2.3$; $p < 0.05$).

### 3.2. Biota and Size Structure

The fish communities showed low taxonomic richness in both groups of islands, with a maximum of four species in a single lake. In the Faroese lakes, the fish community was composed by brown trout *Salmo trutta* Linnaeus 1758, arctic char *Salvelinus alpinus* Linnaeus 1758, Atlantic salmon *Salmo salar* Linnaeus 1758, three-spined stickleback *Gasterosteus aculeatus* Linnaeus 1758, eel *Anguilla anguilla* Linnaeus 1758 and flounder *Platichthys flesus* Linnaeus 1758. In the Azorean lake, the fish community consisted of perch, roach, largemouth black bass, carp, ruivaca, rainbow trout, and pike. The CPUE of the different species is shown in Table S1 and Table S2 in Supplementary Materials.

The size distribution of the sampled fish community, including adults and juveniles, differed between regions (Figure 4A), with a higher frequency of small bodied fish in the warmer Azorean lakes. The median individual size captured in this region was 11.8 cm TL, while it was 18.5 cm TL in the Faroese lakes (Figure 4A). Accordingly, the mean size was significantly higher in the Faroese (21.3 $\pm$ 1.5 TL, SE) than in the Azorean lakes (12.5 $\pm$ 1.4 TL, SE), ($t_{(1,18)}$ = 4.3, $p < 0.001$), When we compared the size of fish relative to the maximum size of resident forms reported in the literature, statistically significant differences remained between regions ($t_{(1,18)}$ = 2.9, $p < 0.05$). Additionally in order to reduce potential confounding factors, such as differences in juveniles habitat use or a particularly strong recruitment year, we compared the size distribution of the sampled fish community including only adults. The mean size of the captured fish community, excluding juveniles from the analysis, was significantly higher in the Faroese (28.2 $\pm$ 1.4 cm TL, SE) than in the Azorean lakes (20.8 $\pm$ 1.7 cm TL, SE), ($t_{(1,18)}$ = 3.4, $p < 0.001$), (Figure 4B). This difference was even higher when comparing the size of the sampled fish community including only adults relative to the maximum size of resident forms reported in the literature ($t_{(1-18)}$ = 7.4; $p < 0.001$).

Total macro-invertebrate taxonomic richness was similar between regions, with 6 $\pm$ 2 and 8 $\pm$ 3 taxa in the Azores and in the Faroe Islands, respectively ($t_{(1,15)}$ = 1.6, $p = 0.13$). There were no differences in the littoral zone ($t_{(1,15)}$ = 0.6, $p = 0.55$), however, richness of macroinvertebrates in the benthic–pelagic zone was higher in the Faroese (4 $\pm$ 1 taxa) than in the Azorean lakes (3 $\pm$ 1 taxa), ($t_{(1,13)}$ = 2.3, $p = 0.04$). Moreover, macroinvertebrate CR from the benthic–pelagic zone was higher in the Faroese lakes (9.4 $\pm$ 6.1) than in the Azorean lakes (2.2 $\pm$ 1.7), ($t_{(1,14)}$ = 2.3, $p = 0.04$), while the CR of littoral macroinvertebrates did not differ significantly ($t_{(1,14)}$ = 1.4, $p = 0.18$).

### 3.3. Potential Predation Pressure on Zooplankton and Cascading Effects

Since fish were significantly smaller in the Azorean than in the Faroese lakes (with and without standardization and with and without juveniles) (Figures 4A,B and 5A,B); and %SF $\times$ PUE was higher in the Azorean lakes ($t_{(1,18)}$ = 4.1, $p < 0.001$) (Figure 6); we would expect a higher fish predation pressure on zooplankton in the Azorean lakes than in the Faroese lakes. This seems to be confirmed. Firstly, zooplankton biomass, mean body size of cladocerans, and of all crustacean zooplankton were larger in the Faroese lakes (Table 3 and Figure 6). Secondly, the total zooplankton:phytoplankton biomass ratio were significantly higher in the Faroese lakes, with no relationship with nutrient concentrations (Figure 3E,F). Thirdly, the log transformed Chl-*a*:nutrient ratios were higher in the Azorean lakes (Table 3 and Figure 6).

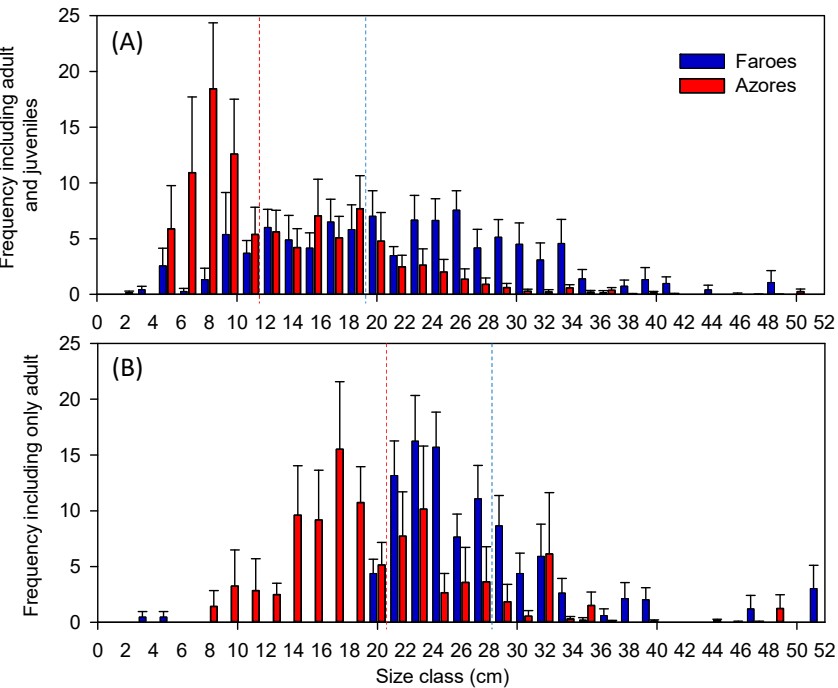

**Figure 4.** Size frequency (cm) of the fish assemblages in each lake from the Azores (red) and the Faroe Islands (blue), in each region, including both adults and juveniles (**A**), and including only adults (**B**), dotted lines represent the median, deviation bars represent SE.

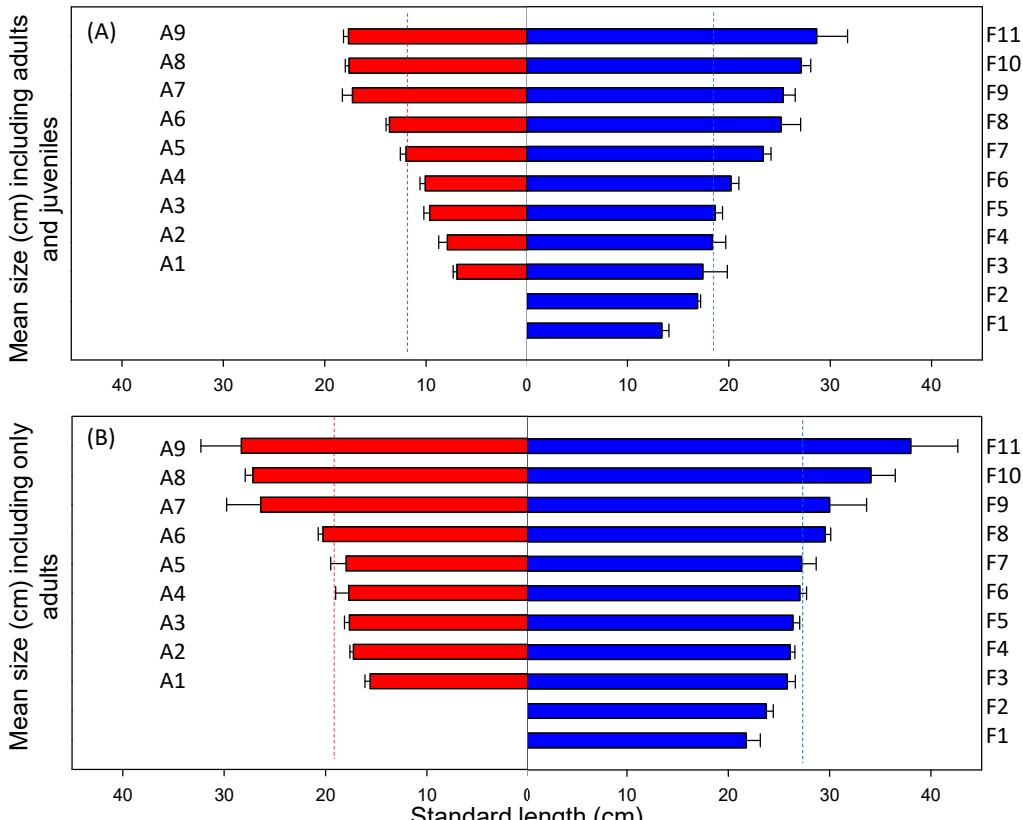

**Figure 5.** Size structure of the fish assemblages (mean total length cm) in lakes in the Azorean (red) and Faroe Islands (blue), including both adults and juveniles (**A**), and including only adults (**B**). Lakes are ordered from high to low average fish size. Lake ID according to Table 1, deviation bars represent SE and dotted lines represent the mean.

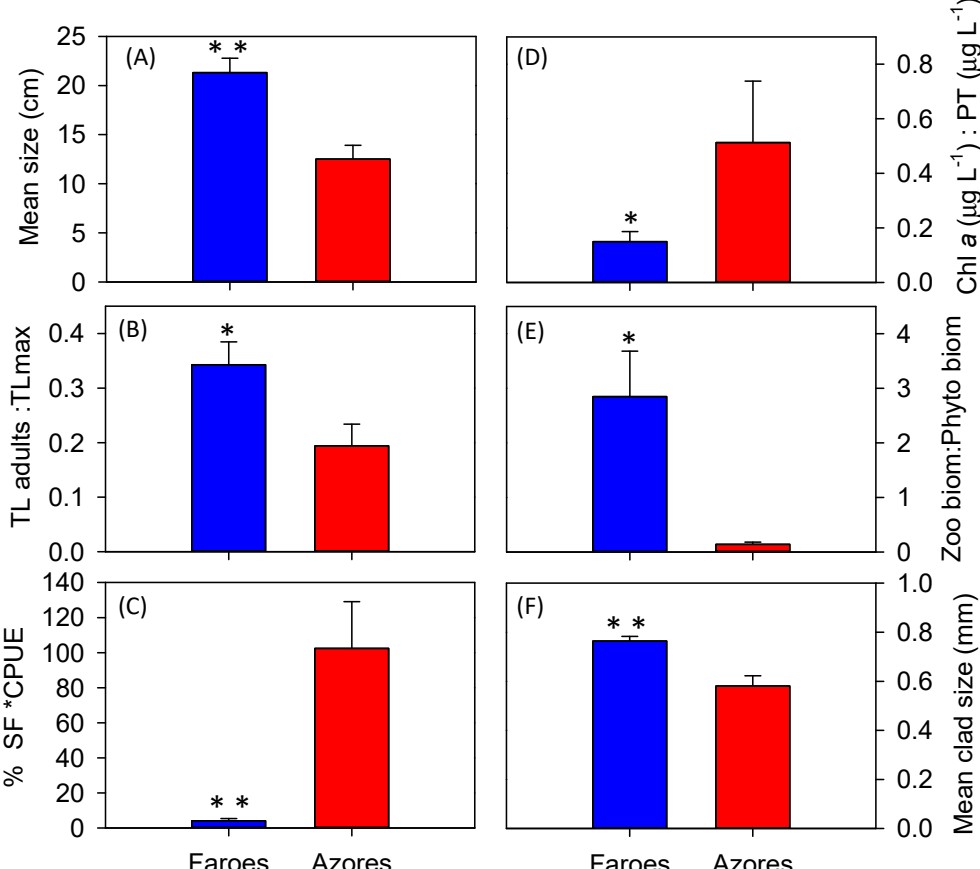

**Figure 6.** Fish community characteristics: mean size (cm) (**A**), mean size standardized by the maximum size reported in literature for each species (TL:Max TL) (**B**), and % of small fish × CPUE (%SF × CPUE) (**C**), and fish predation pressure indicators: chlorophyll-a:total phosphorus ratio (Chl-*a*:TP) (**D**), total zooplankton:phytoplankton biomass ratio (zoo:phytobiomass) (**E**), and mean size of cladocerans (Mean clad size) (**F**). In the Azores (red) and the Faroe Islands (blue); * $p < 0.05$ and ** $p < 0.001$, according to *t*-tests, deviation bars represent SE.

**Table 3.** Fish potential predation pressure indicators, showing mean ± SE values in the two climatic regions and the results of statistical tests (in this case Student *t*-tests).

| | Azores Islands | Faroe Islands | *t* | *p* |
|---|---|---|---|---|
| Total zooplankton mean size (mm) | 0.65 ± 0.05 | 0.82 ± 0.03 | 2.9 | **<0.05** |
| Cladoceran mean size (mm) | 0.58 ± 0.04 | 0.76 ± 0.02 | 4.1 | **<0.001** |
| Total zooplankton biomass (µgL$^{-1}$) | 59 ± 20 | 263 ± 84 | 2.2 | **<0.001** |
| Total zooplankton:phytoplankton biomass ratio | 0.12 ± 0.10 | 2.8 ± 2.6 | 3.1 | **<0.05** |
| Log Chl-*a*:Log TP | 0.64 ± 0.07 | 0.06 ± 0.10 | 4.2 | **<0.001** |
| Log Chl-*a*:Log TN | 0.29 ± 0.05 | 0.06 ± 0.05 | 3.4 | **<0.05** |

Chl-*a*, chlorophyll-*a*; TP, total phosphorus; TN, total nitrogen.

### 3.4. Food Web Metrics

Using the mean position for macro-invertebrates, zooplankton, and fish community in each lake for the two regions, we found a more triangular-shaped food web in the Faroese lakes than in the Azorean lakes (Figure 7). The same trend was detected when analyzing each lake individually, evidenced by the difference between CR invertebrates and fish (base minus top of the food web), which was higher in Faroe Islands than in Azores ($t_{(1–16)} = 4.7$; $p < 0.001$). The food web metrics for the fish community were similar between regions (Table 4). The food web metrics of the basal organisms calculated for each lake were significantly higher in the Faroe Islands, with the exception of TA and SDNNM for which no statistically significant differences were detected (Table 4). In the case of the entire food

web (including fish and basal organisms), the food web metrics had higher values in the Faroese lakes, with the exception of food web length (NR) and centroid distance (CD) and TA, which did not differ significantly (Table 4). Additionally, a significantly lower $\delta^{13}$C signal ($t_{(1-18)}$ = 3.5, $p$ = 0.002), i.e., depleted values were detected in the pelagic zone of the Faroese ($-27.9 \pm 2.7$) than in the Azorean ($-23.4 \pm 3.0$) lakes.

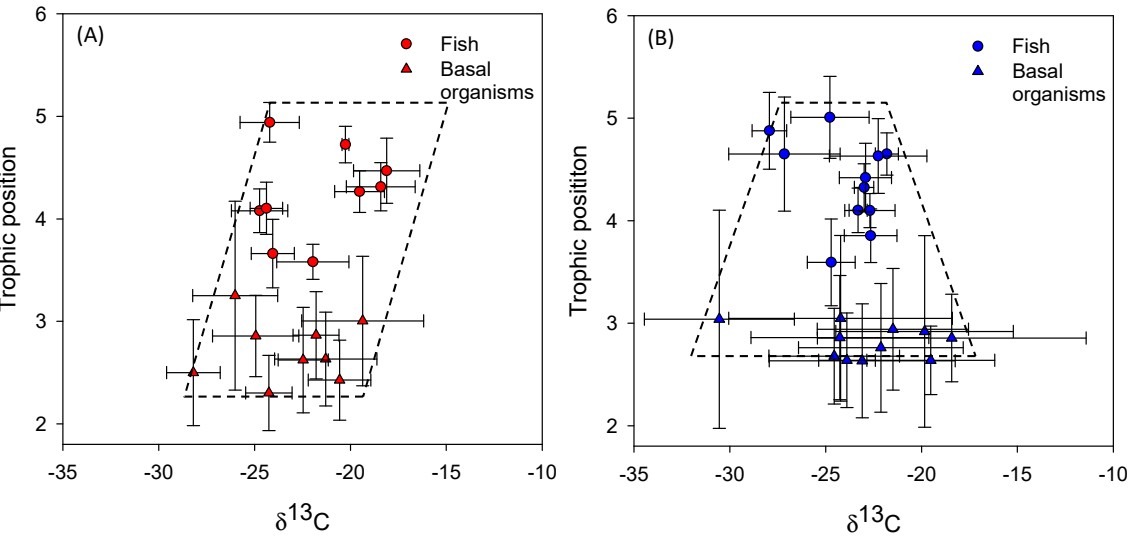

**Figure 7.** General trophic structure of the fish community (circles) and basal organisms (triangles) in lakes in the Azores (red) (**A**) and the Faroe Islands (blue) (**B**). Each point represents the mean in each lake and deviation bars represent SD.

**Table 4.** Comparison of food web metrics at the different trophic levels and for the entire food web in the two regions. The number of lakes used in each analysis is given in brackets. For basal organisms, two lakes were excluded in the Azores Islands due to absence of invertebrates.

| | Fish | | | | Basal Organisms | | | | All Food Web | | | |
|---|---|---|---|---|---|---|---|---|---|---|---|---|
| | Faroe Islands | Azores Islands | $F_{1-18}$ | $p$ | Faroe Islands | Azores Islands | $F_{1-16}$ | $p$ | Faroe Islands | Azores Islands | $F_{1-16}$ | $p$ |
| | (11) | (9) | | | (11) | (7) | | | (11) | (7) | | |
| NR | 0.8 ± 0.44 | 0.7 ± 0.29 | 0.4 | 0.53 | 1.8 ± 0.72 | 1.1 ± 0.37 | **4.5** | **0.05** | 2.9 ± 0.68 | 2.2 ± 0.6 | 2.9 | 0.11 |
| CR | 3.4 ± 2.02 | 2.9 ± 1.4 | 0.3 | 0.59 | 9.0 ± 3.57 | 4.7 ± 2.61 | **7.5** | **0.01** | 9.1 ± 3.75 | 5.1 ± 2.5 | **4.4** | **0.05** |
| CD | 1.0 ± 0.57 | 0.9 ± 0.37 | 0.2 | 0.64 | 2.7 ± 1.04 | 1.3 ± 0.68 | **9.8** | **0.01** | 2.0 ± 0.66 | 1.4 ± 0.6 | 1.57 | 0.23 |
| MNND | 0.4 ± 0.12 | 0.2 ± 0.08 | 6.1 | 0.02 | 1.7 ± 1.12 | 0.6 ± 0.18 | **7** | **0.02** | 0.8 ± 0.21 | 0.4 ± 0.1 | **21.8** | **<0.001** |
| SDNNM | 0.3 ± 0.15 | 0.5 ± 0.11 | 2.7 | 0.12 | 1.3 ± 1.2 | 0.5 ± 0.22 | 3.1 | 0.10 | 0.3 ± 0.45 | 0.3 ± 0.1 | **7.5** | **0.02** |
| TA | 1.9 ± 2.06 | 1.1 ± 0.63 | 1.2 | 0.28 | 7.2 ± 5.79 | 3.2 ± 2.03 | 3.1 | 0.10 | 14.9 ± 9.77 | 6.9 ± 4.2 | 2.8 | 0.11 |
| SEA | 1.5 ± 1.2 | 1.0 ± 0.5 | 1.1 | 0.32 | 7.6 ± 3.8 | 2.9 ± 1.6 | **9.1** | **0.01** | 7.9 ± 3.7 | 3.9 ± 2.0 | **4.8** | **0.04** |
| SEAc | 1.5 ± 1.2 | 1.1 ± 0.5 | 1.1 | 0.3 | 8.4 ± 4.1 | 3.2 ± 1.7 | **10.2** | **0.01** | 8.2 ± 3.8 | 4.0 ± 2.0 | **5** | **0.04** |

NR = nitrogen range; CR = carbon range; CD = centroid distance; MNND = mean nearest neighbor distance; SDNND = standard deviation of the nearest neighbor distance; TA = total area; SEA = standard ellipse area; SEAc = standard ellipse area size corrected, SEAc. F = statistic of ANOVA test, and $p$ represents the significant level. Statistically significant highlighted in bold.

The maximum trophic position of fish did not differ between the two groups of islands ($t_{(1-18)}$ = 1.47; $p$ = 0.16), and neither did the mean trophic position of basal organisms ($t_{(1-18)}$ = 0.93; $p$ = 0.36). The trophic position of fish was positively correlated with the trophic position of basal organisms. This relationship differed between the regions, having a steeper slope in the Faroese (4.1) than in the Azorean lakes (1.3) ($F_{(1-16)}$ = 11.25; $p$ < 0.05) (Figure 8).

Since fish CR can be directly affected by basal CR variability due to intrinsic factors [65], we used the CR of fishes standardized by the CR of the baseline. This ratio represents the

proportion of the CR of basal organisms used by the fish community and was significantly higher in the Azorean lakes ($Z = -1.96$; $p < 0.05$).

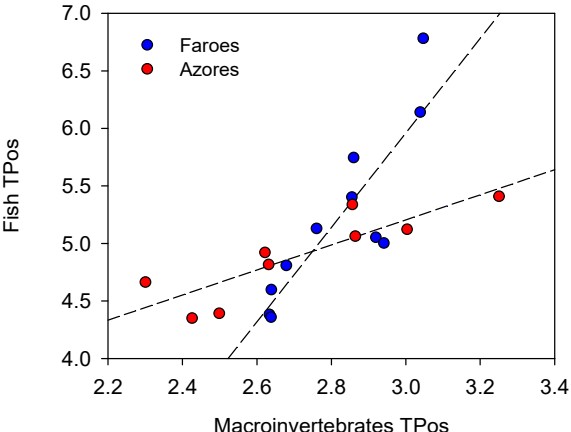

**Figure 8.** Relationship between the trophic position of fish and the trophic position of basal organisms in lakes in the Azores (red) (y = 1.34x + 1.15; $R^2$ = 0.69; $p < 0.05$) and the Faroe Islands (blue) (y = 4.10x − 6.35; $R^2$ = 0.74; $p < 0.05$).

The CR of basal organisms was positively correlated with the morphometric variables area, maximum depth, and Secchi depth in the Faroese lakes, whereas no significant correlations were found for the Azorean lakes (Table 5). Moreover, in the Faroese lakes, the CR of basal organisms was positively correlated with the fish:zooplankton biomass ratio (a measure of fish predation pressure), while no relationship was recorded in the Azorean lakes. Finally, the CR of basal organisms was positively correlated with the CR of primary producers in Faroese lakes, but again no correlation was revealed in the Azorean lakes (Table 6).

**Table 5.** Linear correlation analysis between the environmental variables and CR of basal organisms (log transformed) for the two islands. The number of lakes used in each analysis is given in brackets. For basal organisms, two lakes were excluded in the Azores Islands due to absence of invertebrates.

| | **CR of Basal Organisms** | | | |
| --- | --- | --- | --- | --- |
| | **Faroe Islands (11)** | | **Azores Islands (7)** | |
| | **r** | **p** | **r** | **p** |
| A (ha) | 0.72 | **0.01** | 0.22 | 0.63 |
| Zmax (m) | 0.70 | **0.02** | 0.24 | 0.60 |
| S (m) | 0.71 | **0.02** | 0.64 | 0.25 |
| S:Zmax | −0.52 | 0.10 | 0.27 | 0.66 |
| TN ($\mu gL^{-1}$) | −0.20 | 0.55 | 0.09 | 0.85 |
| TP ($\mu gL^{-1}$) | −0.46 | 0.16 | −0.08 | 0.87 |
| Chl*a* ($\mu gL^{-1}$) | −0.24 | 0.47 | −0.59 | 0.16 |
| Cmax (m) | 0.46 | 0.16 | 0.31 | 0.51 |
| pH | −0.07 | 0.85 | 0.27 | 0.61 |
| Cond | −0.05 | 0.88 | 0.14 | 0.79 |
| $\delta^{13}$C signal of pelagic zone | −0.32 | 0.37 | −0.07 | 0.89 |
| CR primary producers | 0.63 | 0.07 | 0.04 | 0.94 |

See Table 1 for the abbreviations reference. Statistically significant regressions given in bold.

**Table 6.** Linear correlation analysis between the predator pressure indicators (relative abundance of small fish (<10 cm TL) and CR of basal organisms (log transformed) for the two islands. The number of lakes used in each analysis is given in brackets. For basal organisms, two lakes were excluded in the Azores Islands due to absence of invertebrates.

| | CR of Basal Organisms | | | |
| --- | --- | --- | --- | --- |
| | FaroeIslands (11) | | Azores Islands (7) | |
| | r | p | r | p |
| Total zooplankton:phytoplancton biomass ratio | −0.48 | 0.16 | −0.02 | 0.89 |
| Fish:zooplankton biomass ratio | 0.66 | **0.04** | −0.58 | 0.15 |
| % small fish (<10 cm TL) | 0.14 | 0.71 | 0.01 | 0.71 |
| CPUE small fish | −0.05 | 0.88 | −0.13 | 0.91 |

Statistically significant regressions given in bold.

## 4. Discussion

As predicted, we found smaller mean body size and a higher frequency of small-bodied fish in the warmer Azorean region than in the colder Faroese lakes. A random collection of introduced species or random events, for example, a particularly strong recruitment year or differences in life history and habitat use of the juveniles [19] could, potentially, produce such patterns. Additionally, the composition of fish communities differed between regions (i.e., Faroese lakes being dominated by salmonids), life history, and habitat use may vary as well. A larger mean fish size could, potentially, be due to lack of sufficient habitats where juveniles may occur (e.g., streams where the salmonids spawn [19], though likely not the case for the Faroe Islands lakes as they are well connected to streams [46]. However, fish in the warmer Azorean lakes were smaller, even when the size of each fish species was standardized by the maximum body size of resident forms within their continental distribution and when the analysis was performed, excluding juveniles (determined from maturity sizes reported in the literature). The differences in the size structure of the fish communities between the two climatic regions thus followed the pattern obtained along climate gradients or pair-comparisons in continental lakes [16,17,67], as well as within lakes subjected to climate warming [11,68].

Many ectothermic animals experience faster growth at higher temperatures and, consequently, have a smaller body size at the same age than animals subject to lower temperatures (the temperature-size rule, [69,70]. High temperatures often extend beyond the reproduction season of a particular species and enhance fry survival, increasing the abundance of small, young individuals, thus leading to a reduction in the mean size of the fish assemblage [12,15]. This may have contributed likely the smaller-sized fish community structure in the Azores Islands. Other mechanisms, more indirectly linked with higher water temperatures, could contribute to the observed patterns. Small fish usually cope better with low oxygen than their bigger counterparts [71] and particularly a reduced growth under low oxygen levels has been described for salmonids [72]. Consequently, high-temperature driven hypoxia may act as a filter, affecting especially large piscivorous fish, and contributing to the dominance of smaller omnivorous individuals.

Whatever the mechanism, the smaller size of fish in warmer lakes typically leads to a higher predation pressure on zooplankton as showed by [10,25–27]. Supporting our second hypothesis, about stronger cascading effects of fish on zooplankton and phytoplankton in warm lakes, we found a smaller mean size of cladocerans and lower zooplankton:phytoplankton biomass ratio in the Azorean lakes, suggesting the occurrence of lower grazing pressure on phytoplankton compared to the Faroese lakes. This may explain the higher Chl-*a* per unit of TP and TN in the Azores Islands. Other mechanisms, indirectly related to higher temperatures, could also enhance phytoplankton biomass. The near-bottom depletion of oxygen may have promoted phosphorus release from the sediments [73] in the warm, stratified lakes (TP concentration were higher here); whereas, temperature increases as well as reduced body size could lead to an increase in nutrient cycling by the fish community [74,75].

As expected, we found a wider CR for basal organisms in the Faroese lakes. However, contrary to our expectations, the CR of the basal organisms in the Azorean lakes was not related to the proportion of small fish or the fish predation indicators as in the Faroese lakes. Morphometric variables (i.e., area and depth) and transparency (Secchi depth) were positively correlated with the CR of basal organisms in the Faroese lakes. Similarly, lake area and trophic state have been observed to affect the $\delta^{13}C$ signature in the pelagic zone of a wide range of lake areas [41]. This indicates that a combination of biotic and abiotic factors influenced the CR of basal organisms in the colder Faroese lakes. As the Faroese lakes were relatively (i.e., considering water depth) more transparent than the Azorean lakes, a higher benthic production could supposedly be sustained [76,77] in the former. This is supported by the positive relationships between the CR of basal organisms and Secchi disk depth. No such relationships were detected in the Azorean lakes, indicating that in this region other factors may be important in determining the CR of basal organisms. Low dissolved oxygen concentrations at the bottom can negatively affect benthic macroinvertebrates [78,79], often leading to the dominance of a few hypoxic-tolerant species, such as oligochaetes and chironomids [80], with likely lower trophic diversity. Supporting this view, we found a smaller CR and reduced richness of macroinvertebrates in the benthic samples of the Azorean lakes compared to the Faroese lakes.

The $\delta^{13}C$ signal in the pelagic zone is expected to be higher (enriched) in productive [41] and hydrologically stable systems [81], both characteristics stimulated by higher temperatures [82,83]. The potentially higher productivity of the warmer systems may indirectly narrow CR by increasing the pelagic $\delta^{13}C$ signal. Moreover, the $CO_2$ flux between a lake and the atmosphere may be affected by the food web structure [84,85]. A higher phytoplankton biomass due to diminished zooplankton biomass as a consequence of, for example, high fish predation, may enhance the influx of $CO_2$ from the atmosphere and thus enrich the $\delta^{13}C$ signal of the pelagic biota [84].

Despite the differences in CR at the consumer level, no differences were found in fish community CR between the two climatic regions. However, the Azorean fishes showed higher CR when standardized relative to the basal organisms, indicating a greater diversification and a higher efficiency in the use of the resources.

According to the energetic limitation hypothesis (ELH) [29–32], we should expect lower trophic web lengths (TPos and NR) in the warmer regions, but this was not the case in our study. Additionally, a higher proportion of herbivorous fish species is expected in warm lakes [23,24]. Accordingly, the Azorean fish communities were dominated by potentially omnivorous and herbivorous species (up to 50% of abundance the total capture): roach [86] (19%), ruivaca [87] (18%), and common carp [88] (13%). It is accepted that fractionation of nitrogen vary between carnivorous and herbivorous fish, depending on the diet quality (in terms of C:N ratio) and food processing mechanisms [89]. Typically, herbivorous fish exhibit higher per-trophic-level isotope enrichment of nitrogen [90], and consequently, may increase their trophic position more than expected for carnivorous fish. This perhaps explains why we did not find a shorter total food web length in the Azorean lakes. Supporting this view, the fish in the Azorean lakes showed a higher CR relative to the CR of basal organisms, likely reflecting that fish in warmer regions include additional items to the diet such as vegetal material. We cannot, however, fully rule out that the difference in initial fish stock compositions and successions might have affected the food web structure and, thus, the reliability of the comparison of the food web metrics between the two sets of islands.

The space-for-time substitution approach can be criticized as it does not account for biogeographical factors (often leading to high fish species richness at low latitudes) and seasonality differences, thus potentially confounding the effects of climate variability and climate change. The insular lakes used in our study are exposed to the same seasonality in temperature and inhabited by few fish species (some capable of reaching large body sizes) that were introduced from different climatic regions after the 18th century, as in the case of the Azorean lakes. Our data confirmed the dominance of small-sized fishes

in the Azorean lakes, which allows us to cautiously suggest that, in insular lakes, fish assemblages under warm conditions reach smaller sizes. Consequently, we conclude that climate differences, and particularly differences in ambient temperature, have played a key role in the divergence in body size of fish between the two groups of islands leading to different cascading effects among climatic regions. Our results further strengthen the argument that biography is not of key importance for the functional changes occurring in lakes along a temperature gradient. Moreover, our findings suggest that changes in trophic web dynamics and functioning, such as the width at the base of the food web, but not necessarily in the trophic web length, can be expected under current and future warmer climates.

**Supplementary Materials:** The following are available online at https://www.mdpi.com/article/10.3390/w13101380/s1, Table S1: Capture per unit of effort CPUE, refer to average number caught per net per hour of the fish fauna from the nine studied lakes in Azores Islands, Table S2: Capture per unit of effort CPUE refers to the average number of fish caught per net per hour in the 11 studied Faroese lakes. * individuals being caught in fyke nets, Table S3: Total length (cm) measured during this work expressed as mean and SD and maximum (TLMax) and maximum reported in literature (TLMax$_L$) for each species in each region, Table S4: Date of first introduction of each species in Azorean lakes and the origin of the introduction, Table S5: Size of maturation reported in literature for each species in each region used to exclude the juveniles from the analysis. Table S6: Composition of the zooplankton community sampled in Faroese and Azorean lakes.

**Author Contributions:** Conceptualization, N.V., E.J., J.M.N.A., V.G., and M.M.; methodology and investigation, N.V., S.L.A., V.G., J.M.N.A., L.S.J., K.S.C., T.L.L., M.S., R.B., F.L., K.P.B., M.M., and E.J.; data curation and formal analysis, N.V.; writing—original draft preparation, N.V.; writing—review and editing, N.V., S.L.A., V.G., J.M.N.A., L.S.J., K.S.C., T.L.L., M.S., R.B., F.L., K.P.B., M.M., and E.J.; visualization, N.V.; supervision, E.J. and M.M.; project administration, E.J.; funding acquisition, E.J. All authors have read and agreed to the published version of the manuscript.

**Funding:** This research was funded by the Carlsberg Foundation, Research Council of Nature and Universe (272-08-0406), the TK Foundation, Dronning Margrethes og Prins Henriks Fond, Torben and Alice Frimodts Fond, Christian og Ottila Brorsons Rejselegat, Sino-Danish Centre for Education and Research, Aarhus University, University of Copenhagen, the Danish project CRES, and the EU project MARS (Managing Aquatic ecosystems and water Resources under multiple Stress) funded under the 7th EU Framework Programme, Theme 6 (Environment including Climate Change), Contract No.: 603378 (http://www.mars-project.eu, accessed on 30 March 2021). This is a contribution from the Galathea 3 expedition. MM is supported by SNI-ANII Uruguay and the L'Oréal-UNESCO national award for Women in Science and EJ by the TÜBITAK, BIDEB 2232 program (118C250).

**Institutional Review Board Statement:** Not applicable.

**Informed Consent Statement:** Not applicable.

**Data Availability Statement:** Data that are not presented in tables and in the Supplementary Materials are available from the authors on reasonable request.

**Acknowledgments:** We thank the technical staff at Bioscience in Silkeborg for help with the study in the field and lab, and A.M. Poulsen for editorial assistance. The staff at the Department of Biology, University of the Azores, is gratefully acknowledged for fieldwork assistance.

**Conflicts of Interest:** The authors declare no conflict of interest. The funders had no role in the design of the study; in the collection, analyses, or interpretation of data; in the writing of the manuscript, or in the decision to publish the results.

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
