# Peer review of "Food Webs and Fish Size Patterns in Insular Lakes Partially Support Climate-Related Features in Continental Lakes"

_water, doi:10.3390/w13101380_

Round 1
Reviewer 1 Report
Dear Authors,
The comments about your paper is attached below
Good luck for your submission
Best Regards

Reviewer 2 Report
Vidal et al. proposed interesting study deal with lake comparison among two different island which sharing same seasonal pattern with different temperature. In general, I like introduction and discussion. On the other hand, I have doubt about statistical approach which result in problem with data interpretation. However, I would like to mention that after major revision this manuscript should be consider for publication.
Specific comment
line 94-96: This is very old-school statement. Currently is known that trophic discrimination factor can vary a lot in carbon as well as nitrogen. there is chance that these discrimination factor can be affected by species, type of water body, temperature, and etc.
line: 100-105, is it know the effect of habitat richness on this issue?
line 111: citation in missing
line 127-130: are they also Vulcanic?
line 152-165: Here you present a list of species which are present on Azores island? or these lists of species belong to both islands? If so, it would be good to present which species should be present in each island.
line 168-180: please refer table 1 here.
line 169-170: does it mean that you collect samples from the deepest zone or you collected samples pelagial area above deepest point.
line 191: also, no connection with your supplement. I would suggest linking your tables with your text more properly.
line 197-198: which dept were used how many nets per each depth were used?
line 205-217: have you collect detritus? It can be also important basal food source.
line 242-245: but it mean you do not include lake identity as a factor or random effect, right? So, you just merge data from distinct region? If so, I suggest re-done this analysis and include lake identity as random effect or as factor. Both give you important outputs.
line 247-250: Again, I have doubt about this approach. It seems that you underestimated the effect lake identity. Which can be another major player. I would suggest that in most of your analysis where you compare lakes, this component should be included. So might be something like GLM should be applied.
257-264 I would definitely say that you need to change your statistical approach and added suitable models which can help you analyze such differences. might be GLM or better some Linear mixed model with effect
line 274-275: again lake identity is important parameter.
line 277-286: I am wondering about this package. I thought is already gone and should not be used because some errors can appear- At least, this was highly suggested in last ISOECOL 2018, in Chile. When you did your data analysis If this or last year, I would rather suggest used package which are currently supported like SIBER or MixSIAR.
Results: In general, although results are nicely written and are clear, I have problem to believe given statistic, as not cover lake identity which might interact with region specificity. Thus, even your results can be correct, its definitely needed to change your point of view on this problem
line 317-318: but it might be given by lake identity and might be their interaction with regions
linw 333: Table 1- when I see this data, I have to say that lakes are pretty different thus your stat should changed
line 433-435: „significantly lower δ13C“ its mean depleted ?
line 520-523: I am not really sure by meaning positively corelated? So it’s mean that with increasing transparency, CR is more wider?
line 553-555. in terms of previous sentence this statement can be true. However, you should consider and discuss that initial fish stock were different between both island. So, then fish stock succession might be also differ. Thus, suggested theory from line 550-553 can be misleading at this case.
line 560-563: or it might be also caused by different species composition and their food source utilization and competition as well as amount of free niche across all ecosystems.
Author Response
May 4, 2021
Dear Guest Editor Water
We would like to thank you for your fast response and the two reviewers for their valuable comments. We are sure that the manuscript has improved, thanks to the reviewer´s comments.
We carefully revised the manuscript as suggested by the reviewers; in particular we improved the use of the terminology (i.e. Faroe Island and Faroese lakes) as was suggested by the second reviewer. We also moved the paragraph including the information about the fish introduction from materials and methods to the introduction section suggested by both reviewers.
We hereby submit the revised version of the manuscript. Please find below our point-by-point responses (in red) to the reviewers’ comments and suggestions. In the manuscript you can find the changes highlighted in red.
Thank you in advance for your attention to this new version of the manuscript
Best regards,
Dr. Nicolás Vidal
Vidal et al. proposed interesting study deal with lake comparison among two different island which sharing same seasonal pattern with different temperature. In general, I like introduction and discussion. On the other hand, I have doubt about statistical approach which result in problem with data interpretation. However, I would like to mention that after major revision this manuscript should be consider for publication.
Specific comment
line 94-96: This is very old-school statement. Currently is known that trophic discrimination factor can vary a lot in carbon as well as nitrogen. there is chance that these discrimination factor can be affected by species, type of water body, temperature, and etc.
Answer to reviewer: Point taken. We modified the sentence in the new version of the manuscript please see page 2 lines 95-98: “The carbon signal (δ13C) was thought to be transmitted with little change from the primary producers to the consumers; consequently the consumers´ signal depends directly on the signal of their diet [38]. However, large variability in trophic fractionations has been reported for aquatic consumers…”
line: 100-105, is it know the effect of habitat richness on this issue?
Answer to reviewer: we included examples that are directly related with climate. Habitat richness may also play a role, but we have no information of such effect it in this stud. So therefore was not included in the introduction.
line 111: citation in missing
Answer to reviewer: we included a reference see page 3 line 114
line 127-130: are they also Vulcanic?
Answer to reviewer: Yes are also volcanic, we complemented the information in the new version of the manuscript. See page 4 line 153 “The Faroe Islands are a treeless archipelago also of volcanic origin composed of 18 islands located in the North Atlantic Ocean”
line 152-165: Here you present a list of species which are present on Azores island? or these lists of species belong to both islands? If so, it would be good to present which species should be present in each island.
Answer to reviewer: We now moved this section to the introduction see page 3 lines 123-135. The information in this section is for the Azores; in the case of the Faroese lakes the species were transplanted internally and less information is available. The complete list of species in each system is now included in the supplementary material.
line 168-180: please refer table 1 here.
Answer to reviewer: We refer to table 1 in the paragraph; see page 5 line 177 of the new version of the manuscript.
line 169-170: does it mean that you collect samples from the deepest zone or you collected samples pelagial area above deepest point.
Answer to reviewer: The second, we integrated several water samples from different depths and we use this water for the subsequent analysis. We included a sentence indicating how the deepest point of each lake was determined see page 5 lines 184-186: “The deepest point of each lake was determined using a portable echo sounder (Hondex ps-7, Toyohashi, Japan).”
line 191: also, no connection with your supplement. I would suggest linking your tables with your text more properly.
Answer to reviewer: we don´t see the need to link this sentence with supplement: “Fish were captured using Lundgren gillnets with 14 mesh sizes ranging from 6.25 to 75.0 mm from knot to knot as well as by fyke nets to supplement the capture of species that are not easily caught in gillnets (e.g. eel, Anguilla spp.).”
line 197-198: which dept were used how many nets per each depth were used?
Answer to reviewer: Thanks for asking. The text is now changed to: The gillnets were placed overnight in the littoral and pelagic zone. If shallow (mean depth<3m) only benthic nets were used and half of the nets were set in each zone; if mean depth was >3m 1/3 of the nets were set as for shallow lakes and an additional 1/3 of the nets were pelagic nets positioned in the middle of the epilimnion. Please see page 6 line 206-210
line 205-217: have you collect detritus? It can be also important basal food source.
Answer to reviewer: we collected sediment samples containing detritus (0.5 cm slice of sediment cores collected using a Kajak corer). Please see page 6 line 220
line 242-245: but it mean you do not include lake identity as a factor or random effect, right? So, you just merge data from distinct region? If so, I suggest re-done this analysis and include lake identity as random effect or as factor. Both give you important outputs.
Answer to reviewer: answered below
line 247-250: Again, I have doubt about this approach. It seems that you underestimated the effect lake identity. Which can be another major player. I would suggest that in most of your analysis where you compare lakes, this component should be included. So might be something like GLM should be applied.
Answer to reviewer: answered below
257-264 I would definitely say that you need to change your statistical approach and added suitable models which can help you analyze such differences. might be GLM or better some Linear mixed model with effect
Answer to reviewer: answered below
line 274-275: again lake identity is important parameter.
Answer to reviewer: We resume here the answer for the three previous concern from the reviewer, we agree with the reviewer that other statistical approaches (e.g., models selection) are possible, but this does not mean, in our opinion, that the current approach is wrong or invalid. In regards to include lake effect as a random factor looks promising; however, for our response variables we only have one data in each lake e.g. layman metrics, zooplankton biomass, mean size of fish, etc. in our opinion is not possible to include the lake as a factor if only one data per lake.
line 277-286: I am wondering about this package. I thought is already gone and should not be used because some errors can appear- At least, this was highly suggested in last ISOECOL 2018, in Chile. When you did your data analysis If this or last year, I would rather suggest used package which are currently supported like SIBER or MixSIAR.
Answer to reviewer: We agree with this suggestion, we recalculated the metrics using package SIBER, please see page 7 line 290 in the method section. Also we modified the result in table 4 and in page 13 lines 443-447.
Results: In general, although results are nicely written and are clear, I have problem to believe given statistic, as not cover lake identity which might interact with region specificity. Thus, even your results can be correct, its definitely needed to change your point of view on this problem
Answer to reviewer: please see the previous answer on this issue
line 317-318: but it might be given by lake identity and might be their interaction with regions
Answer to reviewer: please see the previous answer on this issue
linw 333: Table 1- when I see this data, I have to say that lakes are pretty different thus your stat should changed
Answer to reviewer: please see the previous answer on this issue
line 433-435: „significantly lower δ13C“ its mean depleted ?
Answer to reviewer: yes depleted, we modified the sentence “Additionally, a significantly lower δ13C signal (t(1-18)=3.5, p=0.002), i.e. depleted values were detected” please see page 13 line 448
line 520-523: I am not really sure by meaning positively corelated? So it’s mean that with increasing transparency, CR is more wider?
Answer to reviewer: yes the result showed a positive correlation between CR of basal resources and Secchi depth.
line 553-555. in terms of previous sentence this statement can be true. However, you should consider and discuss that initial fish stock were different between both island. So, then fish stock succession might be also differ. Thus, suggested theory from line 550-553 can be misleading at this case.
Answer to reviewer: see next comment please
line 560-563: or it might be also caused by different species composition and their food source utilization and competition as well as amount of free niche across all ecosystems.
Answer to reviewer: Point taken. We now write: We cannot, however, fully rule out that the difference in initial fish stock compositions and successions might have affected the food web structure and thus the reliability of the comparison of the food web metrics among the two sets of Islands. Please see page 18 lines 584-587

Reviewer 3 Report
The manuscript is devoted to the comparison of freshwater reservoirs of the Faroe Islands and the Azores in terms of numerous biotic parameters. The work impressed me as a comprehensive study worthy of publication. The comments are given below.
Lines 130-139 and Figure 2: this belongs to the Introduction section.
Lines 151-166: this belongs to the Introduction section
Like 169-170: specify what is the deepest point of each lake (m) or link to the Table 1.
Line 173: add manufacturer and country
Line 192: add reference to the construction of this fyke net
Line 198: add reference to the appropriate fish density estimation method
Lines 257-261: this belongs to Discussion
Line 269: why in particular you analyze cladocerans among plankton animals?
Lines 272-274: did you use any software to calculate ANCOVA and other indices?
Lines 315-321, Table 1: A, lake area and Zmax, maximum depth were measured by you? If not, this information should be in the Introduction or in M&M.
Table 2. Add abbreviations to the footnote so the reader did not have to look for it through the paper
Lines 356-360: add full species name at the first mention of species. There is no need in parentheses.
For example: …sea trout, Salmo trutta Linnaeus 1758, Arctic char, Salvelinus alpinus (Linnaeus 1758), …
Check www.fishbase.se for common and Latin names.
Line 385 and elsewhere: The Faroe Islands, or simply Faroes. Be consistent.
Table 4: add Z to the footnote
Round 2
Reviewer 2 Report
I carefully read answers of Authors. I have to say I agree with their answers, they did good job. Unfortunately, they have just one record per lake thus I understand that using linear model with mixed effect can be tricky or invalid. This is a pity situation; on the other hand I fully understand that multiple lake sampling in both region is demanding. Hopefully, there will some future study whose recycle this dataset together with their own sampling camping and bring more light in hidden lake processes.
Nevertheless, I satisfied and recommend this manuscript for publication